# Design of Low RCS Circularly Polarized Patch Antenna Array Using Metasurface for CNSS Adaptive Antenna Applications

**DOI:** 10.3390/ma12121898

**Published:** 2019-06-13

**Authors:** Jianxing Li, Tayyab A. Khan, Juan Chen, Muhammad U. Raza, Anxue Zhang

**Affiliations:** 1School of Electronic and Information Engineering, Xi’an Jiaotong University, Xi’an 710049, China; jianxingli.china@xjtu.edu.cn (J.L.); chen.juan.0201@xjtu.edu.cn (J.C.); engr.usmanraza@gmail.com (M.U.R.); anxuezhang@xjtu.edu.cn (A.Z.); 2Department of Electronic Engineering, City University of Hong Kong, Hong Kong SAR 999077, China

**Keywords:** metamaterial, metasurface, circular polarization, patch antenna array

## Abstract

A low radar cross section (RCS) circularly polarized patch antenna array operating at the downlink S-band (2492 ± 5 MHz) of the Chinese Compass Navigation Satellite System (CNSS) is proposed. The low RCS is achieved by replacing the conventional metallic ground with an artificial magnetic conductor (AMC)-based metasurface. Two different AMC unit cells are designed having a phase difference within 180 ± 37° and combined in a chessboard-like configuration to realize the AMC-based metasurface. Furthermore, the AMC-based metasurface is utilized as the ground of the CNSS array for wideband RCS reduction. A wideband RCS reduction from 6 GHz to 17 GHz is achieved due to the wideband diffusion property of the AMC unit cells. The maximum RCS reduction is more than 14 dB at 13.3 GHz irrespective of the polarization direction of the incident waves. Moreover, the circular polarization (CP) performance is realized by embedding a circular slot on the patch radiator of the antenna element. The radiation characteristics of the CNSS array are hardly impacted by the inclusion of the metasurface-based ground. The proposed CNSS array has been fabricated and measured. The measurement results are in reasonable agreement with the simulations. The proposed CNSS array can be a good candidate for CNSS adaptive antenna applications where low RCS is simultaneously demanded.

## 1. Introduction

In the past decade, the manipulation of waves at will has received tremendous interest, underpinned by the advent of the transformation method [1] and metamaterials [2,3]. A wide variety of novel devices, such as invisibility cloaks and advanced lenses, have been realized in electromagnetics and other fields. Nowadays, many researchers are working to broaden their horizons for the practical applications of metamaterials. In this work, a metasurface is employed to achieve the low scattering property of a circularly polarized antenna array that is applicable to the Chinese Navigation Satellite System (CNSS) adaptive antennas, where low radar cross section (RCS) is simultaneously demanded.

Satellite navigation receivers are susceptible to signal noise and intentional jamming [4]. To alleviate this issue, adaptive anti-jam antennas based on antenna arrays are generally employed. On the receiver side, these arrays should work efficiently to cater to the weak satellite signals on the earth. A great deal of effort has been exerted to propose different antenna arrays for the Global Navigation Satellite System (GNSS) [5,6,7]. However, almost all of the GNSS antenna array works are focused on the design considerations without investigating the inevitable issue of high RCS due to the larger aperture of the array.

According to the radar range equation [8], the radar detection range is directly proportional to the RCS of the object. The stealth platforms require the RCS of the antenna array to be relatively small; therefore, it is necessary to reduce the RCS of such GNSS arrays. Different techniques have been proposed in the literature to reduce the antenna scattering, such as object shaping method [9], radar absorbing material [10], and active/passive cancellation [11,12]. The limitations of these methods are the narrow bandwidth and their effect on the radiation characteristics of the antenna. Apart from these, a number of absorption methods have also been proposed such as the Salisbury screen [13], electromagnetic band gap (EBG) combined with lumped resistance [14], and perfect metamaterial absorber [15,16]. However, these methods also suffer from narrowband RCS reduction. Therefore, these techniques are not suitable for applications where wideband RCS reduction is needed. Later, the combined properties of a perfect electric conductor (PEC) and artificial magnetic conductor (AMC) were utilized for backscattered energy cancellation [17]. The in-phase reflection of AMC restricts the band of RCS reduction. To overcome this issue, different solutions have been proposed, including polarization reflective surfaces (PRS) and frequency selective surfaces (FSS) [18,19,20,21,22,23]. Basically, the phase difference between the two AMCs is controlled to meet the criterion of phase cancellation. In this way, effective cancellation can be realized over broadband. However, the problem with these types of structures is that they considerably increase the complexity of the overall system. Most of these works are also focused on in-band RCS reduction, while out-of-band RCS also plays a key role in radar detection. The out-of-band RCS reduction of a planar antenna using absorbing ground plane was achieved in [24]. However, due to the implementation of three different structures, the overall design is complicated. Other than RCS reduction, the gain enhancement [25] and polarization reconfiguration [26] of different antennas, based on metasurface concepts, have been presented in previous studies to depict some unique applications of the metamaterials.

This paper proposes a new design to realize the out-of-band RCS reduction of a CNSS array using metasurface. In order to realize wideband RCS reduction, two different AMC unit cells with a phase difference within 180 ± 37° are designed and then combined in a chessboard-like configuration to function as a ground plane of the CNSS array. The diffusion property of the proposed AMC unit cells helps to minimize scattering of the antenna surface from 6 GHz to 17 GHz. Simulation and measurement results confirm the wideband RCS reduction without affecting the radiation characteristics of the CNSS array.

## 2. Analysis and Design of Low RCS CNSS Antenna Array

A tremendous amount of research has been carried out in the literature to design the antenna arrays for GNSS applications. For the stealth environment, these antenna arrays are the scattering source of electromagnetic waves due to their metallic structure. The available prototypes in the literature only discuss the design of these arrays. Considering the importance of low scattering in a stealth environment, a novel design of the CNSS array with low RCS property in out-of-band is achieved in this paper. Our design of the CNSS array has some benefits: Firstly, it tackles the RCS issue of the antenna array for the navigation applications. Secondly, the structure of the design is simple and easy to be fabricated. Thirdly, the RCS from 6 GHz to 17 GHz is significantly reduced which is beneficial to stealth platforms where the CNSS adaptive antenna is a necessity.

### 2.1. Design of the Metasurface as a Ground of CNSS Array

A metasurface comprising of different AMC unit cells helps to realize RCS reduction [17]. The property of RCS reduction depends on the phase cancellation behavior of the unit cells, which further yields by the phase difference between the unit cells. The RCS reduction upon incidence of a plane wave to the metasurface with different AMC unit cells can be approximated by Equation (1), where θ_1_, θ_2_, B_1_, and B_2_ are the reflection phases and reflection coefficients of unit cell 1 and unit cell 2, respectively. According to Equation (1), if the phase difference between the unit cells is equal to 180°, the RCS reduction will be maximized.
(1)$$\sigma (dBsm)=10log\left[{\left|\frac{B_{1}e^{j\theta_{1}}+B_{2}e^{j\theta_{2}}}{2}\right|}^{2}\right]$$

However, as the reflection phase of the unit cells varies with frequency, the ideal phase difference of 180° is difficult to maintain over a wide frequency band. Therefore, compared with the same sized PEC, the 10 dB RCS reduction is usually regarded as a criterion for the RCS reduction. According to that, the effective phase difference of 180 ± 37° between the unit cells can produce 10 dB RCS reduction.

To meet the criterion of RCS reduction, two different AMC unit cells are designed. The geometries of the proposed AMC unit cells are shown in Figure 1. The structures of the unit cells consist of metallic patches, etched on Taconic TLX-6 substrate with a thickness of 1.52 mm, a dielectric constant of 2.65, and a tangent loss of 0.0019. Both the unit cells are backed by a metallic ground plane. The optimized dimensions of unit cells are also listed in Figure 1. All the simulations are carried out in the frequency domain solver of CST Microwave Studio (Dassault Systèmes, Vélizy-Villacoublay, France). The parametric optimizer in CST is used to obtain the optimized dimensions of the unit cells. The periodicity of both the unit cells is the same to ensure the same number of periods of each block in a chessboard-like arrangement. The simulation model of unit cell 1 in the CST software 2017 can be seen in Figure 2a. The Floquet ports are used along Z and −Z directions to excite the unit cell. In the X and Y directions, the unit cell boundary conditions are applied to simulate the unit cell model. In our design, the metasurface would be applied as the ground of the CNSS antenna array. Therefore, it is necessary to study the effect of antenna substrate on the reflection phases and magnitudes of the AMC unit cells. The Taconic TLX-6 (Taconic Biosciences, Rensselaer, NY, USA) is used as the antenna substrate with a thickness of 3.18 mm. Figure 2b shows the reflection phase and magnitude of both unit cells without the effect of the upper antenna substrate. It can be seen that the reflection from the surface of the unit cells is in different phases under the normal incidence of plane waves and the reflection magnitude nearly equals to 0 dB, indicating that almost all the energy is reflected without being absorbed. Figure 2c depicts the reflection phases and magnitudes of both the unit cells with the effect of the upper antenna substrate. It further shows that the reflection phases have shifted to lower frequencies due to the effect of the extra substrate. However, the reflection magnitude of both the unit cells is approximately equal to 0 dB. It can be noted from Figure 2b that an effective phase difference of 180 ± 37° is obtained between the unit cells from 8 GHz to 16 GHz. After the analysis and design of the AMC unit cells, block arrays consisting of 5 × 5 AMC of unit cell 1 and unit cell 2 are designed to satisfy the periodic boundary conditions. The size of the array ground is selected to be large enough to keep the ideal performance (that is obtained using periodic boundary). These arrays are further placed in a chessboard-like configuration to scatter the incoming energy into different directions. The design of the full metasurface consisting of unit cells can be seen in Figure 3.

### 2.2. Design of the Proposed CNSS Array

The design configuration and geometrical model of the proposed CNSS array along with the AMC-based metasurface are demonstrated in Figure 4. The array consists of four radiating patches etched on an antenna substrate backed by the AMC-based metasurface. The inter-element spacing between elements in both directions is kept as 0.4λ_0_ at 2492 MHz (where λ_0_ is the free-space wavelength) to produce good isolation. According to [5], the right-hand circular polarization (RHCP) is required for the CNSS operations. The axial ratio (AR) is a parameter that describes the purity of the circular polarization. If the value of the AR is 0 dB, the antenna is considered as perfectly circularly polarized. Usually, the AR of below 3 dB is applied as criteria to define a circularly polarized antenna. The effect of different geometrical shapes on the AR performance of the antenna has been investigated in [27]. Based on [27], the circular slots are cut on the right bottom corners of the patch elements at the location of (X,Y) = (9.1, −9.1) to achieve circular polarization (CP). The radius of these circular slots is 6.85 mm. The two-layer structure is illustrated in Figure 4. The proposed AMC-based metasurface is applied under the CNSS array to achieve the wideband RCS reduction between 6 GHz and 17 GHz. However, it is important to ratify the influence of this metasurface on the radiation performance of the CNSS array. The value of the phase difference among the unit cells is approximately zero at CNSS S-band (2491 ± 5 MHz); therefore, it is expected that the AMC ground would not affect the radiation performance of the CNSS array. Before evaluating the radiation and scattering performance of the proposed CNSS array, the CNSS array with a metallic ground plane is taken as a reference array to compare the results with the proposed CNSS array. The structure of the reference CNSS array consists of the radiating patches with circular slots backed by the metallic ground. The simulated performance of the reference CNSS array with the metallic ground is depicted in Figure 5a,c,e. The return loss bandwidth of the antenna array elements is from 2.4 GHz to 2.6 GHz (relative bandwidth of about 8%). Figure 5a depicts that the resonance frequency of the antenna elements is 2.492 GHz. Due to the 0.4λ_0_ spacing between the array elements, good isolation performance can be seen in Figure 5c. The parametric study is performed by changing the radius of the circular slots to produce good AR. Figure 5e shows that the AR is less than 3 dB and the antenna array elements exhibit good CP performance at the designed frequency band.

### 2.3. Scattering and Radiation Performance of the Proposed CNSS Array

To assess the performance of the designed CNSS array, scattering and radiation performance are simulated. Figure 5 compares the simulated performance of the reference and proposed CNSS antenna array. Figure 5b,d,f shows the reflection coefficients, AR, and transmission coefficients performance of the proposed CNSS array. The reflection coefficient, AR, and isolation of the array elements are less than −15 dB, 3 dB, and −20 dB, respectively, with the CNSS S-band. The plausible agreement between the simulated performance of the reference and proposed antenna array can be observed from Figure 5. The simulated monostatic RCS of the proposed CNSS array compares to the same sized metallic plate as a reference is presented in Figure 6. The simulations are carried out with the aid of the time domain solver of the CST software. For military and navigation applications, the transmitting and receiving antennas are far from the target surface. Therefore, the incident wave can be considered equivalent to the plane wave. In simulations, plane waves impinging from the normal direction to the surface of the proposed CNSS array are considered to study the RCS. It can be noticed that the monostatic RCS of the designed CNSS array in the band (6–17 GHz) has a dramatic reduction with the comparable metallic plate, when the plane waves normally impinged on the surface of the CNSS array. This RCS reduction is mainly owing to the effective phase difference of the unit cells, which dispersed the incidence energy to the other directions and a lower amount of energy retransmitted by the surface to the incidence direction. It can be observed that the RCS reduction band is from 6 GHz to 17 GHz (a relative bandwidth of 96%). The 10 dB RCS reduction is from 8.2 GHz to 14.8 GHz (a relative bandwidth of 58%). The proposed structure also has good polarization stability for the x- and y-polarized incident plane waves. The maximum RCS reduction of more than 14 dB can be observed at 13.3 GHz.

The metasurface with a random arrangement of the AMC unit cells can also be utilized as the ground of a CNSS array instead of the chessboard configuration. The overall diffused pattern from the surface of the CNSS array would be different between the random and chessboard arrangements of the unit cells. Whether better or worse RCS reduction performance can be obtained for a random arrangement, as compared with the chessboard arrangement, depends on the random placement of the unit cells and the overall diffused patterns owing to these random arrangements of the unit cells. The random arrangement of the unit cells changes in each iteration of the simulation. Therefore, RCS reduction would be different in each iteration. The optimal solution to the random placement of the unit cells and the RCS performance can be obtained using optimization algorithms such as genetic algorithm (GA) and particle swarm optimization (PSO). The RCS performance of the proposed antenna array with a random combination of the AMC is further investigated to study the effect of random placement of the unit cells on the RCS. The AMC unit cells arranged in a random arrangement can be seen in Figure 7a. The comparison of the RCS reduction with the chessboard configuration and random placement of the unit cells is given in Figure 7b. It can be seen from Figure 7b that the chessboard metasurface has a wide 10 dB relative bandwidth as compared to random surface. However, the maximum RCS reduction can be seen in the case of random arrangement. Due to the wide 10 dB RCS reduction bandwidth, the metasurface with the chessboard configuration is further considered for the investigation of the low RCS antenna array design.

Figure 8 depicts the 3-D scattering patterns comparison between the proposed CNSS array with chessboard configuration and a metallic ground of the same size. It can be noticed from Figure 8a that the scattering energy from the surface of the metallic ground is strongly reflected back to the incident direction mainly in the single lobe. However, for the case of the proposed CNSS array in Figure 8b, the incidence energy is diffused into many lobes at φ = 45°, 135°, 225°, and 315°, resulting in a low RCS property of the CNSS array. Figure 9a,b shows the bistatic RCS performance of the proposed CNSS array for an x-polarized incident wave in comparison with a metallic ground at 12.2 GHz. Due to the effect of the AMC-based ground, effective RCS reduction can be observed. In the xoz plane, the RCS of the designed CNSS array is effectively diminished in the angular domain of −35° ≤ θ ≤ −20°, −15° ≤ θ ≤ +15°, and 20° ≤ θ ≤ 60°. In the yoz plane, the RCS reduction can be realized in between ±30°. Figure 10 represents the normalized E-plane (φ = 0°, θ = −180°–180°) and H-plane (φ = 90°, θ = −180°–180°) CP radiation patterns at 2492 MHz. Pure RHCP performance is obtained at 2492 MHz with good CP isolation. Simulated results verify the radiation performance of the designed CNSS antenna array design at 2492 ± 5 MHz.

## 3. Fabrication and Measurements

The proposed CNSS array is fabricated and presented with the measurement setup in Figure 11. The proposed CNSS array design consists of the antenna and the metasurface layer. The antenna layer is composed of CP radiation patches while the metasurface comprises of the AMC unit cells. The metasurface is glued under antenna array to act as a ground for the proposed CNSS array. The thickness of the glue is 0.08 mm with the dielectric constant of 4.4 and a loss tangent of 0.02. All the measurements are conducted inside the anechoic chamber. An Agilent E8363B network analyzer (Agilent Technologies, Santa Clara, CA, USA) is used to measure the return loss and isolation of the fabricated prototype. Figure 12 shows the reflection coefficients, AR, and transmission coefficients performance of the proposed CNSS array. The resonant frequencies are slightly shifted to the lower frequencies. However, the measured 10 dB return loss bandwidths of the designed array elements cover the frequency range of 2.4–2.6 GHz. Moreover, the AR and isolation between the array elements are less than 3 dB and 20 dB, respectively, across the CNSS S-band. Figure 13 demonstrates the measured radiation patterns at 2492 MHz. Good concurrence among the simulated and measured results can be seen as the proposed array shows good RHCP performance. Therefore, it can be concluded that the radiation performance of the proposed CNSS antenna array has not affected by the inclusion of the metasurface as a ground of the antenna array.

Limited by the experimental conditions, only the reflectivity is measured to attain the RCS reduction of the proposed CNSS array in the monostatic configuration, as depicted in Figure 14. The obvious RCS reduction can be seen from 6 GHz to 17 GHz frequency band for both the simulated and measured results. The proposed CNSS array has a dramatic RCS reduction as compares with the metallic ground of the same size. It can be observed that the RCS reduction is independent of the polarization behavior of the incident waves. The significant mismatch concerning the simulated and measured results of the RCS of the proposed CNSS array are more likely because of the fabrication tolerances and measurement errors. Particularly, the glue, which is used to attain a single structure from the two layers, will also have some effect on the measurements. However, the RCS reduction values for the measurement results are showing some peaks from 9 to 11 GHz and from 13 to 15 GHz. The measured results show significantly better results compared to the simulated results, which is more likely due to the measurement errors. For the RCS performance of the proposed CNSS surface, the reflectivity of the surface is measured for the normal incidence of the plane waves. For the monostatic RCS, the transmitter and receiver should be at the same location with the direct line of sight to the surface. The peaks from 9 to 11 GHz, 13 to 15 GHz shows that the receiver was not exactly present in front of the proposed CNSS surface to collect the reflections from the surface. Due to that error, the receiver picked less received signals. However, the RCS reduction is obvious throughout the band of 6 GHz to 17 GHz.

Table 1 tabulates the comparison of the proposed work and some of the recent studies on the metasurface and metamaterials. The key functionality of the proposed CNSS array is its low scattering property. Moreover, the radiation efficiency and gain of the proposed antenna array is 96 % and 8 dBi, respectively, at the designed frequency band.

## 4. Conclusions

A novel design for the CP antenna array working in CNSS S-band (2492 ± 5 MHz) with low RCS and good radiation performance is presented in this paper. Low scattering properties are realized by loading the novel metasurface containing different AMC unit cells as a ground of the CNSS array. Owing to the effective phase difference of 180 ± 37° between the unit cells, the scattered energy is redirected into different directions causing RCS reduction from 6 GHz to 17 GHz band, with maximum RCS reduction greater than 14 dB at 13.3 GHz. Furthermore, due to the different working frequency of the CNSS array and metasurface, the radiation performance of the CNSS array is hardly affected. Therefore, good impedance matching, isolation, and pure RHCP performance of the CNSS array can be observed in simulated and measured results. 

## Figures and Tables

**Figure 1 materials-12-01898-f001:**
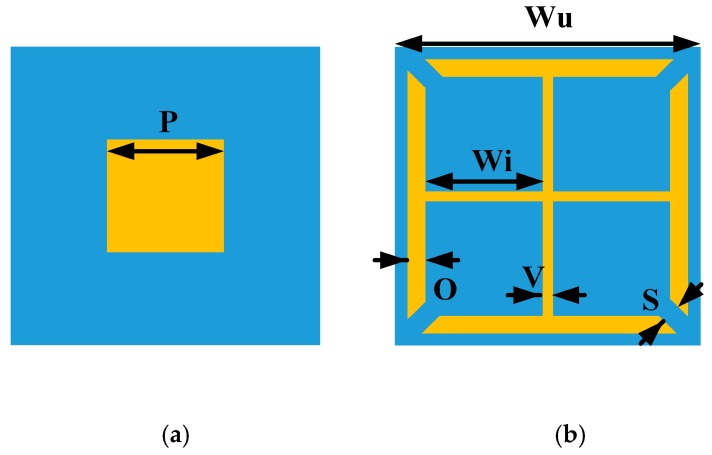
The artificial magnetic conductor (AMC) unit cells with the following dimensions: Wu= 10 mm, P = 5 mm, V = 0.2 mm, Wi = 3.9 mm, O =0.8 mm, S = 0.4 mm. (**a**) Unit cell 1. (**b**) Unit cell 2.

**Figure 2 materials-12-01898-f002:**
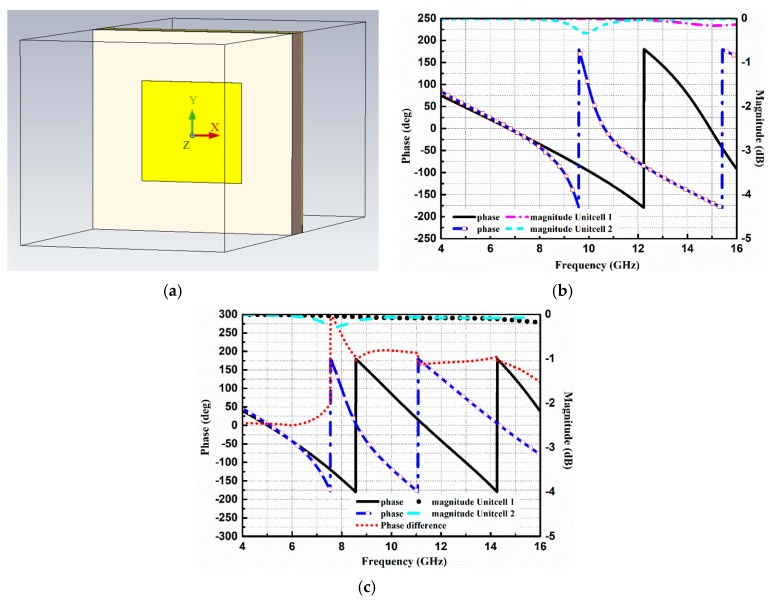
Reflection performance of the AMC unit cells: (**a**) Simulation model of unit cell 1. (**b**) Without the antenna substrate. (**c**) With the antenna substrate.

**Figure 3 materials-12-01898-f003:**
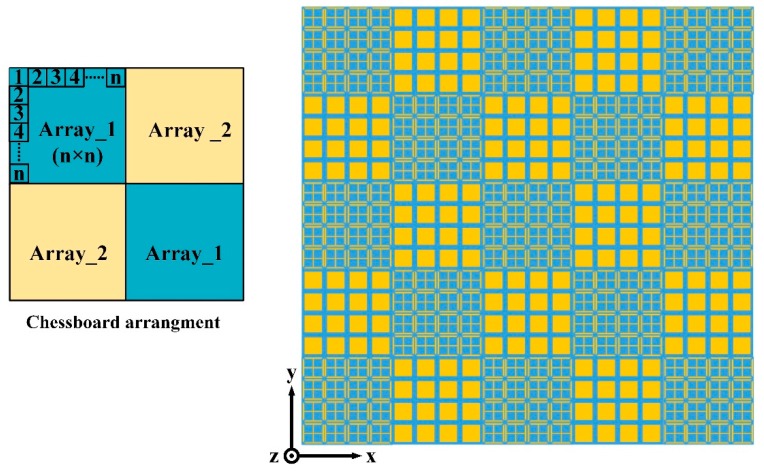
Geometry of the metasurface with AMC unit cells in a chessboard-like configuration.

**Figure 4 materials-12-01898-f004:**
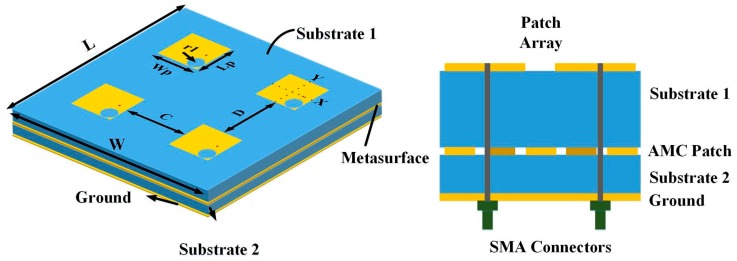
Configuration of the proposed Chinese Navigation Satellite System (CNSS) array with L = W = 200 mm, Wp = Lp = 32 mm, r1 = 6.85 mm, C = D = 47.8 mm.

**Figure 5 materials-12-01898-f005:**
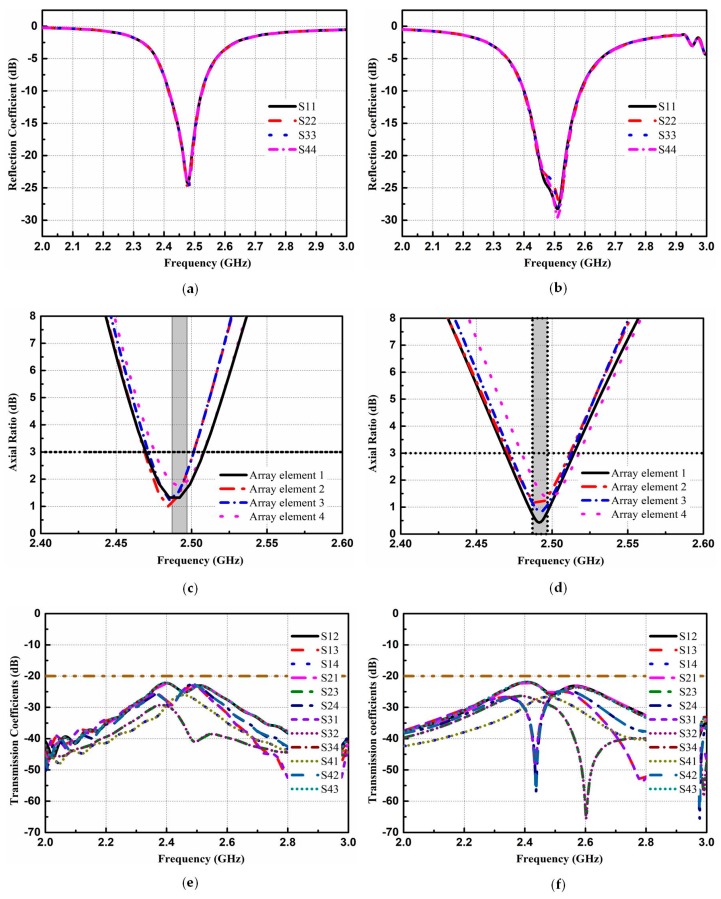
Results of the CNSS antenna array with a metallic ground (reference) and a metasurface (proposed): (**a**) Return loss (reference). (**b**) Return loss (proposed). (**c**) Axial ratio (AR)(reference). (**d**) AR (proposed). (**e**) Isolation (reference). (**f**) Isolation (proposed).

**Figure 6 materials-12-01898-f006:**
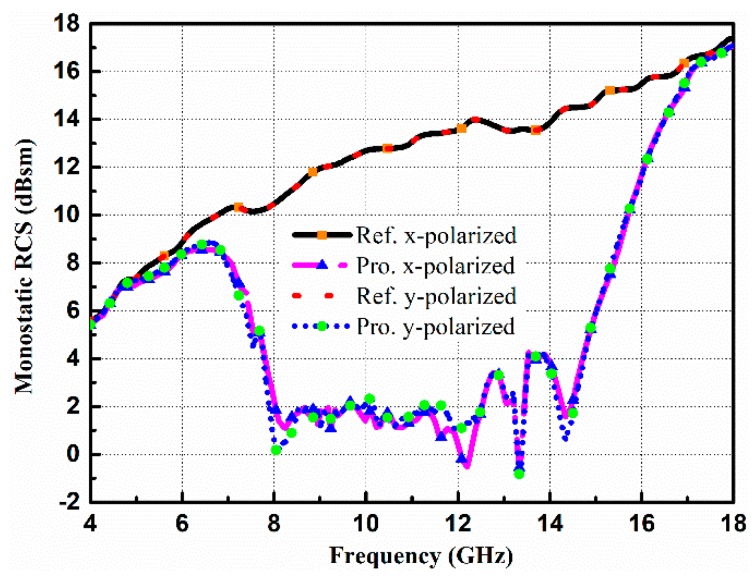
Monostatic radar cross section (RCS) performance of the proposed CNSS array.

**Figure 7 materials-12-01898-f007:**
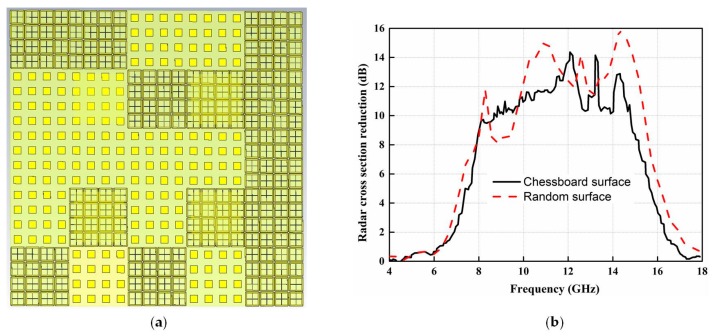
The metasurface with a random arrangement of the unit cells: (**a**) Random arrangement. (**b**) RCS reduction comparison.

**Figure 8 materials-12-01898-f008:**
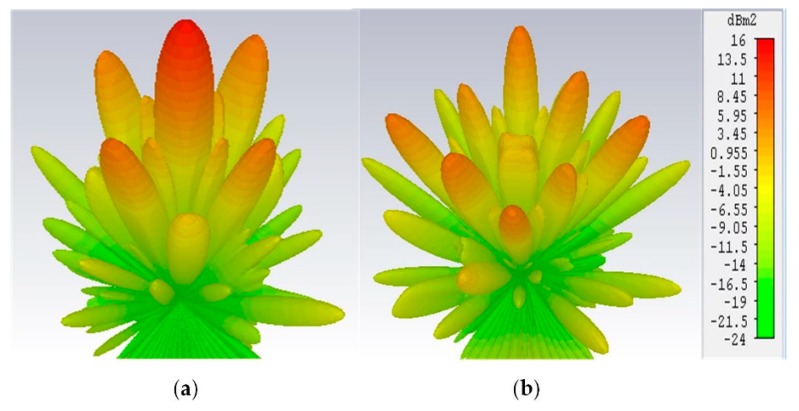
Simulated 3D scattering patterns at 13.3 GHz: (**a**) Metallic plate. (**b**) Proposed CNSS array.

**Figure 9 materials-12-01898-f009:**
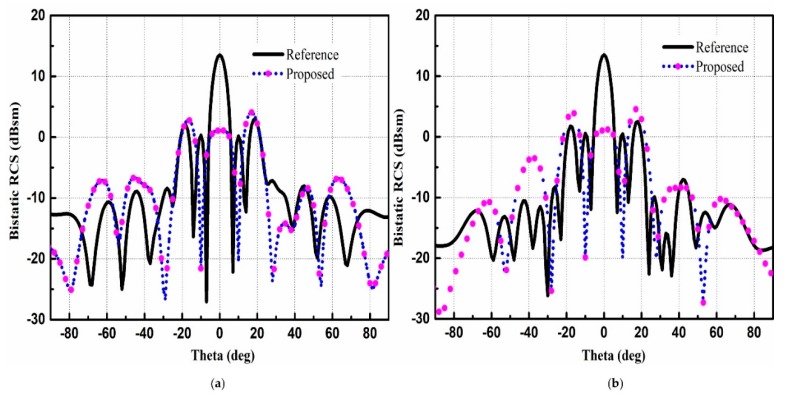
Bistatic RCS pattern analysis: (**a**) xoz plane. (**b**) yoz plane.

**Figure 10 materials-12-01898-f010:**
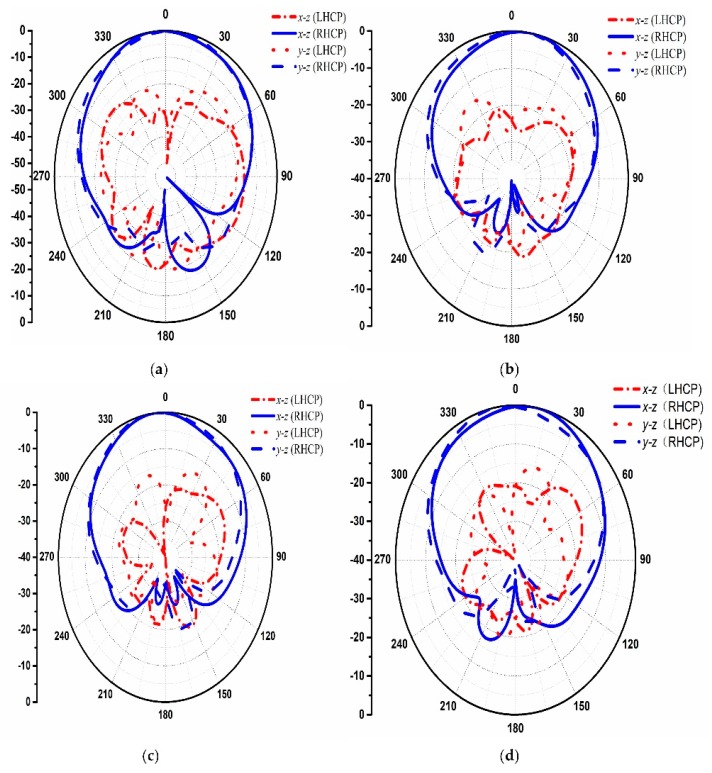
Simulated circular polarization (CP) radiation patterns at 2492 MHz: (**a**) Element 1. (**b**) Element 2. (**c**) Element 3. (**d**) Element 4.

**Figure 11 materials-12-01898-f011:**
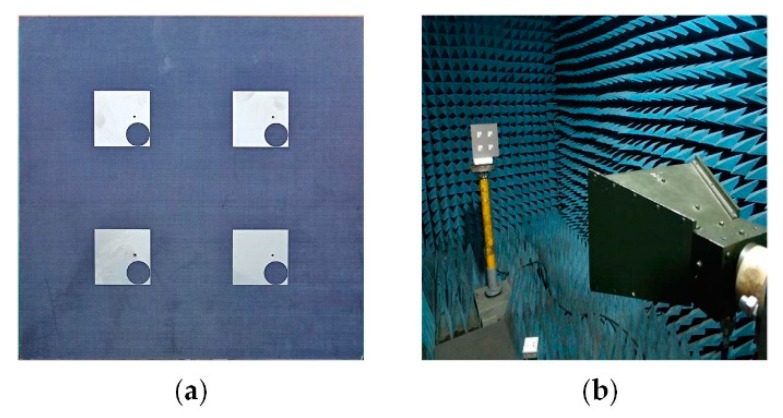
Prototype of the proposed CNSS array: (**a**) Fabricated array. (**b**) Test environment.

**Figure 12 materials-12-01898-f012:**
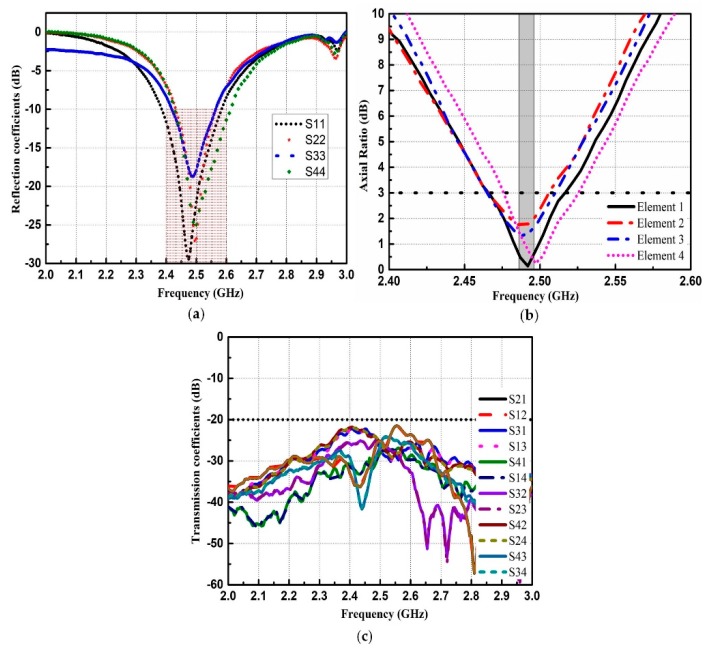
Measured results of the proposed CNSS antenna array: (**a**) Return loss. (**b**) AR. (**c**) Isolation.

**Figure 13 materials-12-01898-f013:**
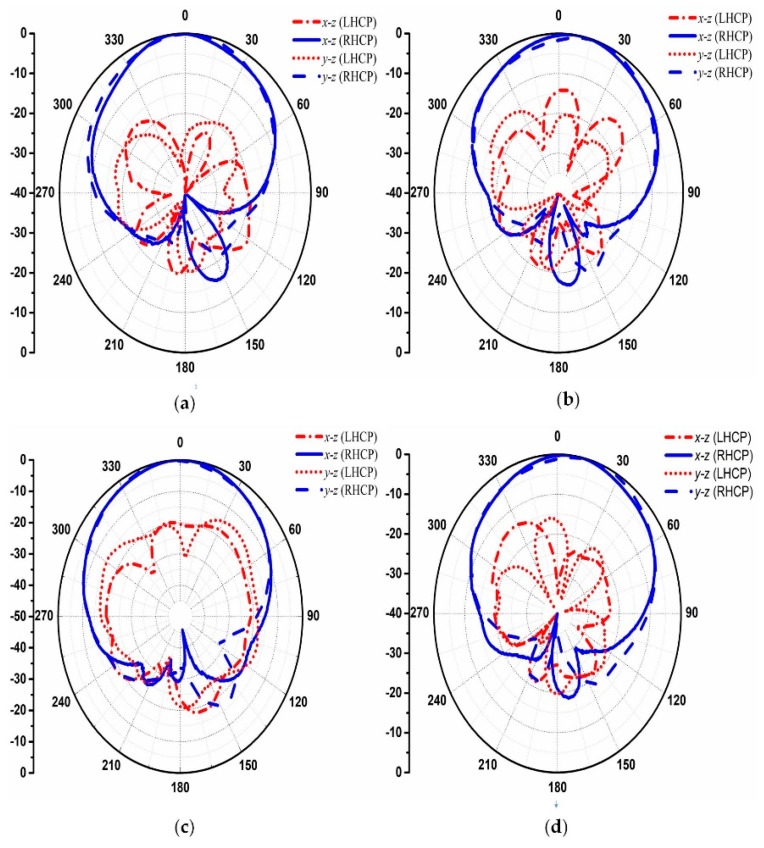
Normalized measured radiation patterns at 2492 MHz: (**a**) Element 1. (**b**) Element 2. (**c**) Element 3. (**d**) Element 4.

**Figure 14 materials-12-01898-f014:**
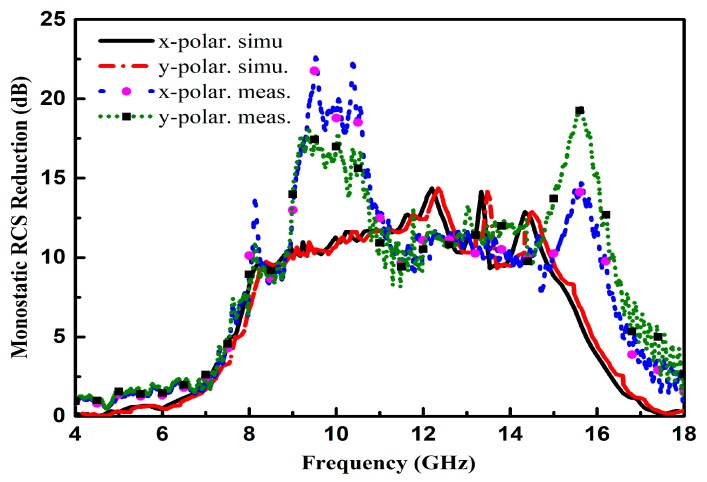
Simulated and measured RCS reduction comparison for both polarizations.

**Table 1 materials-12-01898-t001:** Comparison between this work and some recent metasurface antennas.

Reference	Bandwidth (%)	Gain (dBi)	Efficiency	Functionality
[28]	6.0%	6.0	NG	Reconfigurable
[29]	16%	15.5	73%	High gain
[30]	110%	3.0	60%	Ultra wideband
Proposed	8%	8.0	96%	RCS reduction

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
