# Peer review of "Design of Low RCS Circularly Polarized Patch Antenna Array Using Metasurface for CNSS Adaptive Antenna Applications"

_materials, 2019, doi:10.3390/ma12121898_

Reviewer 1 Report

Review report for “Design of Low RCS Circularly Polarized Patch Antenna Array Using Metasurface for CNSS Adaptive Antenna Applications” by J. Li et al.

The authors propose to replace a patch antenna’s ground plane with a checkerboard metasurface consisting of artificial magnetic conductors in order to achieve broadband radar-cross-section reduction of the antenna. Numerical and experimental results are presented.

General/conceptual comments:

·The authors state in the introduction that they achieve “wideband RCS reduction without affecting the radiation characteristics of the CNSS array” yet they do not present reference simulations and/or experiments with a conventional metallic ground plane to support this claim. Only the sentence “The phase difference between the unit cells is approximately zero at CNSS-S band (2491 ± 5MHz), therefore, it is expected that AMC ground would not affect the radiation performance of the CNSS array. ” follows on that topic. The comparison in terms of the radiation properties of the antenna (directivity, return loss, ..) with their AMC ground plane and with a conventional metallic ground plane is very important but lacking. The authors should at least add a numerical investigation thereof, comparing results as presented in Fig. 9 with the reference case of using a metallic ground plane.

·What happens if instead of a regular checkerboard as in Fig. 4 a random combination of the two types of AMC was chosen? In other words, how important is the order of the checkerboard? A simple numerical investigation could answer this question.

Detailed comments:

·Lines 75-76: “if the phase difference between the unit cells is equal to 180°, the scatter energy from the surface to the incidence will vanish 76” this is only true if B1=B2. Is this condition fulfilled in practice? If yes, then the authors should use B instead of B1 and B2. If no, they should correct that statement.

·Lines 89-90: “optimized dimensions” The authors should provide details about the optimization procedure they used. The reader should be able to understand how they obtained the designs presented in Fig. 1. If the authors deem this a digression in the main text, the details can be added in an appendix.

·Fig. 2 (and subsequent figures): the legend should clarify which curve is magnitude and which curve is phase. A sketch of the considered setup should also be added.

·Why a 5x5 chessboard configuration? What is the reason for using 5x5 rather than, say, 1x1 or 20x20? The reader must be able to understand this choice.

·Lines 120-121: details of the geometry of the circular slots are required. Moreover, the choice of its geometrical parameters has to be explained or a reference should be provided.

·Figure 6: An equation defining how RCS was calculated is necessary in the main text. Over what surface is the reflected signal integrated? What illumination is used? These details are important to ensure the presented results can be reproduced.

·Line 161 and Fig. 8: what does the “o” in “xoz” plane stand for?

·Figure 9: a definition of “axial ratio” in the main text is necessary.

·Lines 192-193: “Especially, the glue, which is used to attain the single structure from the two layers, will also have some effect on the measurements. ” What glue was used? This detail is important, in particular if the authors believe it impacted their results.

·Fig 14: there is an error in the figure legend, two curves are labelled as “y-polar. simu.”.

·Lines 200-201: “discrepancies between the simulated and measured results” The significant mismatch between experiment and simulation requires a thorough discussion to be explained. The provided sentences does not provide any insight. Is the RCS reduction in the experiment for 9-11 GHz and 13-15 GHz significantly better than expected from simulations? Why? 

Overall, I think this manuscript merits publication in mdpi Materials once my concerns have been fully addressed.

Author Response

Thank you very much for your comments on our manuscript. 

Reviewer 2 Report

1) Application(s) of the proposed antenna should be better detailed in the abstract.

2) Why the structure is named "a metasurface antenna"? Please explain it in more details.

3) Please highlight the novelty of the proposed antenna in the section 2 as analysis and design section.

4) Please merge figures 2 and 3 for better comparison.

5) the Following relevant references are suggested to be added (in the introduction section).

- D. M. Pozar, Microwave engineering, John Wiley & Sons, 2009.

- “Metasurface antenna with switchable polarization”, 2015 International Conference on Communications and Signal Processing (ICCSP), 2-4 April 2015, Melmaruvathur, India.

- "Metamaterial electromagnetic energy harvester with near unity efficiency", Applied Physics Letters, vol. 106, no. 15, pp. 153902, 2015.

- “A true metasurface antenna”, 2016 IEEE International Symposium on Antennas and Propagation (APSURSI), 26 June-1 July 2016, Fajardo, Puerto Rico.

6) To increase the validity of your work please add a comparison section and compare the performance parameters such as dimension, bandwidth, radiation gain, and radiation efficiency of the proposed work with recent papers based on metasurfaces and metamaterials. After that, please list the results in a table. I have found some papers based on metamaterial and metasurfaces, as listed below.

- “Design of a PIN Diode-Based Reconfigurable Metasurface Antenna for Beam Switching Applications”, International Journal of Antennas and Propagation, Volume 2019, Article ID 7216324, 7 pages, https://doi.org/10.1155/2019/7216324.

- “Design of a wide-gain-bandwidth metasurface antenna at terahertz frequency”, AIP Advances 7, 055313 (2017); https://doi.org/10.1063/1.4984274.

- “Metamaterial-Based Antennas”, Proceedings of the IEEE, Volume: 100, Issue: 7, Page(s): 2271 – 2285, July 2012.

- “A Negative Index Metamaterial to Enhance the Performance of Miniaturized UWB Antenna for Microwave Imaging Applications”, Appl. Sci. 2017, 7, 1149; doi:10.3390/app7111149.

- "Periodic FDTD analysis of leaky-wave structures and applications to the analysis of negative-refractive-index leaky-wave antennas", IEEE Trans. Microw. Theory Tech., vol. 54, no. 4, pp. 1619-1630, Jun. 2006.

- "CRLH traveling-wave and resonant metamaterial antennas", IEEE Antennas Propag. Mag., vol. 50, no. 5, pp. 25-39, Oct. 2008.

Author Response

Thank you very much for your comments on our manuscript. 

Round  2

Reviewer 1 Report

The authors made a considerable effort to address all concerns and questions raised by the reviewers which I appreciate. I have the following further minor suggestions that I trust the authors will implement. Then the manuscript is ready for publication.

A few minor recommendations:

·For easy comparison, I suggest to merge Figs. 5 and 9: use 2 columns and 3 rows, 1 column for the current Fig. 5 (the reference), and one column for the current Fig. 9 (the proposed checkerboard). Then the first row compares return loss, the second AR and the third the isolation.

·The authors’ made a preliminary investigation on random vs checkerboard organization of the proposed ground plane. They should include these interesting results in the manuscript, rather than only mentioning them in their reply to my comments.

·“After the analysis and designing of AMC unit cells, block arrays consisting of 5 × 5 AMC of unit cell 1 and unit cell 2 are designed to satisfy the periodic boundary conditions.“ I’m not sure I understand this sentence. Do the authors mean they need an n × n AMC of unit cell 1 and unit cell 2 with n being an odd number to satisfy the periodic boundary conditions, and they picked 5 as example of an odd n? Please rewrite the sentence to be clearer regarding why you chose n=5.

·“The receiver was not exactly present in front of the proposed CNSS surface to collect the reflections from the surface. Due to that error, the receiver picked less received signals” I tried to understand this issue by looking at Fig. 11b but am not sure I understand it. Do you evaluate RCS based on the reflection signal measured with the horn antenna? If yes, why don’t you place the horn antenna directly in front of the CNSS surface to avoid this problem? Please explain exactly what measured data you use to calculate RCS in the experiment.

Author Response

Thank you for your good comments.
